# The Soybean Laccase Gene Family: Evolution and Possible Roles in Plant Defense and Stem Strength Selection

**DOI:** 10.3390/genes10090701

**Published:** 2019-09-11

**Authors:** Quan Wang, Guang Li, Kaijie Zheng, Xiaobin Zhu, Jingjing Ma, Dongmei Wang, Kuanqiang Tang, Xingxing Feng, Jiantian Leng, Hui Yu, Suxin Yang, Xianzhong Feng

**Affiliations:** 1Key Laboratory of Soybean Molecular Design Breeding, Northeast Institute of Geography and Agroecology, Chinese Academy of Sciences, Changchun 130102, China; wangquan@iga.ac.cn (Q.W.); liguang@iga.ac.cn (G.L.); zhengkaijie@iga.ac.cn (K.Z.); majingjing@iga.ac.cn (J.M.); wangdongmei@iga.ac.cn (D.W.); tangkuanqiang@126.com (K.T.); fengxingxing112@163.com (X.F.); lengjiantian@iga.ac.cn (J.L.); yuhui@iga.ac.cn (H.Y.); yangsuxin@iga.ac.cn (S.Y.); 2School of Resources and Environment, University of Chinese Academy of Sciences, Beijing 100049, China; 3School of Life Science, Jilin Agricultural University, Changchun 130118, China; ca_zhuxiaobin@163.com

**Keywords:** laccase, Cu-oxidase, *GmLac*, resistance, selection, soybean, stem strength

## Abstract

Laccase is a widely used industrial oxidase for food processing, dye synthesis, paper making, and pollution remediation. At present, laccases used by industries come mainly from fungi. Plants contain numerous genes encoding laccase enzymes that show properties which are distinct from that of the fungal laccases. These plant-specific laccases may have better potential for industrial purposes. The aim of this work was to conduct a genome-wide search for the soybean laccase genes and analyze their characteristics and specific functions. A total of 93 putative laccase genes (*GmLac*) were identified from the soybean genome. All 93 GmLac enzymes contain three typical Cu-oxidase domains, and they were classified into five groups based on phylogenetic analysis. Although adjacent members on the tree showed highly similar exon/intron organization and motif composition, there were differences among the members within a class for both conserved and differentiated functions. Based on the expression patterns, some members of laccase were expressed in specific tissues/organs, while some exhibited a constitutive expression pattern. Analysis of the transcriptome revealed that some laccase genes might be involved in providing resistance to oomycetes. Analysis of the selective pressures acting on the laccase gene family in the process of soybean domestication revealed that 10 genes could have been under artificial selection during the domestication process. Four of these genes may have contributed to the transition of the soft and thin stem of wild soybean species into strong, thick, and erect stems of the cultivated soybean species. Our study provides a foundation for future functional studies of the soybean laccase gene family.

## 1. Introduction

Laccases, *p*-diphenol:oxygen oxidoreductases (EC 1.10.3.2), represent the largest subgroup of multicopper-containing oxidases (MCOs), also known as “blue” oxidases [1,2,3]. They are widespread not only in fungi and plants, but also in bacteria and insects [4,5,6,7]. Laccase was initially discovered in Japanese lacquer tree by Yoshida in 1883 [8]. Subsequently, it was defined as a metal-containing oxidase [9]. Commonly, the laccase protein sequence has three conserved domains (Cu-oxidase, Cu-oxidase_2, and Cu-oxidase_3) that are used to identify canonical laccases. The active center of laccases contains four copper atoms: a type-1 copper, a type-2 copper, and two type-3 copper atoms, which are responsible for electron transfer in the catalytic process [10,11]. The catalytic capacity of laccases is actually nonspecific. In most cases, laccase substrates are aromatic compounds; however, it also has the ability to catalytically oxidize non-aromatic compounds [12,13,14,15]. Precisely, because of their nonspecific catalytic ability, laccases were reported to be involved in various physiological activities in living organisms and are utilized in diverse biotechnological applications in industrial production.

Laccases used to be classified as fungal, plant, or bacterial laccases [2,16]. Basic characteristics and functions of laccases are diverse both within and across kingdoms. Delignification is probably the most studied function of fungal laccases [17], as it can be identified in nearly all wood-decomposing fungi [18], although the detailed mechanism of this process is still unclear [19,20]. In addition to the delignification function, fungal laccases are also connected with many other physiological processes including the regulation of morphology, such as fruiting body formation [21,22,23], pathogenicity when the fungi are plant pathogens [24,25], promotion of special compound synthesis and metabolism such as pigment formation [26], and flavonoid metabolism [27].

Unlike fungal laccases, plant laccases have a higher glycosylation degree, which is closely related to secretion, thermal stability, and enzyme activity [28]. The isoelectric point (pI) of fungal laccase is around 4.0, whereas the isoelectric point of plant laccases is usually higher than this value (usually above 5.0) [28,29]. These observations suggest that plant and fungal laccases may have functional differences. In plants, laccases are reported to play a role in forming protective barriers. For example, the secretion of laccases in lacquer tree helps wound healing to prevent infection [30,31]. Plant laccases were also reported to regulate plant organ development [32]. In *Arabidopsis*, *SKU5* encoding a laccase was localized to both the plasma membrane and the cell wall. Loss of *SKU5* function exhibited a left-handed rotation bias in roots and etiolated hypocotyls [33]. Plant laccases were suggested to contribute toward pathogen defense [34,35]. The most well-known function of plant laccases is probably the involvement of the enzyme in the lignification process [36,37]. For example, an increased lignin level was observed in transgenic poplar plants following over-expression of the cotton laccase gene *GaLAC1* [38].

Although the presence of bacterial laccase is widely recognized [4,39], reports of characterized and purified bacterial laccase are still very limited. There are relatively few studies on the function of bacterial laccases. Unlike plant and fungal laccases, which are often extracellular, bacterial laccases are intracellular [5]. A well-studied bacterial laccase is CotA, identified in *Bacillus subtilis* and suggested to participate in the biosynthesis of the brown spore pigment to prevent ultraviolet (UV) damage [40]. Laccases were also identified in some marine bacteria. For example, Lac15 identified from the South China Sea provides chloride tolerance and dye discoloration ability to the bacteria and may have specific industrial applications [41]. In general, laccase is a widely distributed enzyme with some functional differentiation across the kingdoms, providing diversity to laccases for industrial usages. In addition, laccase was always a topic of study ever since it was discovered because of its wide range of potential functions for the fields of environmental protection, biological detection, paper making, and food industries. Thus, identification of novel laccases is always meaningful to facilitate efficient industrial production.

In plants, although laccases were identified and classified in many species such as *Arabidopsis thaliana* [3,20], sorghum [42], rice [43], *Populus* [28], and cotton [44], an investigation of the laccase gene family in soybean is yet to be conducted. In this study, we identified all possible laccase-coding genes from the soybean reference genome (Wm82.a2.v1). We then predicted the gene and protein structures, studied the expression patterns of the gene family with or without *Phytophthora sojae* infection, and investigated the selection pressure exerted on the gene family during the domestication process. This study also generates a strong base for future soybean laccase research.

## 2. Materials and Methods

### 2.1. Identification of Laccase Gene Family Members in Soybean

Soybean genome was downloaded from Phytozome (https://phytozome.jgi.doe.gov). The genome of *Arabidopsis* was obtained from The *Arabidopsis* Information Resource (TAIR) database (http://arabidopsis.org). In total, 17 laccase sequences were obtained according to published research [3]. *Arabidopsis* laccase members contain Cu-oxidase, Cu-oxidase_2, and Cu-oxidase_3 (PF00394, PF07731, and PF07732) domains. The three domains were searched in the soybean genome using HAMMER 3.0. We took the intersection of the three BLAST results and manually eliminated genes with no open reading frame and multi-transcripts from the same gene. Additionally, identified genes were verified using the SMART domain database (http://smart.embl-heidelberg.de).

### 2.2. Physical Map of the Soybean Laccase Genes and Properties of the Laccase Proteins

Gene identifiers (IDs) and chromosomal locations were obtained from the Phytozome database (https://phytozome.jgi.doe.gov/). Mapchart 2.32 was used to visualize laccase gene (*GmLac*) distribution on soybean chromosomes [45]. Peptide size (amino acids, aa), isoelectric point (pI), and protein molecular weight (Da) were calculated using the online ExPASy tool (https://web.expasy.org/protparam/) [46]. Subcellular locations of the *GmLac* members were determined using the online software CELLO (http://cello.life.nctu.edu.tw/) [47]. Transmembrane topology and signal peptides were predicted using the online Phobius software (http://phobius.sbc.su.se/index.html) [48].

### 2.3. Sequence Alignment and Phylogenetic Analysis

An alignment of 93 GmLac and 17 AtLac peptides was performed using ClustalW in MEGA 7.0 [49]. All conserved sites were used to construct evolutionary relationships with the neighbor-joining (NJ) method (1000 replicates). The final phylogenetic tree was visualized and edited in iTOL (http://itol.embl.de/) [50].

### 2.4. Gene Duplication and Expansion Analysis

Intra-species synteny blocks were detected by the MCScanX algorithm [51], with following settings (five genes required to call a collinear block based on the previous all-to-all BLASTP result E-value ≤ 1 × 10^−5^). The ratio of nonsynonymous substitutions per nonsynonymous site (Ka) to synonymous substitutions per synonymous site (Ks) was computed using KaKs_Calculator 2.0 [52]. Then, the estimated date (mya, million years ago) of each duplication event was calculated using the mean Ks values (T = Ks/2λ), assuming clocklike rates (λ) of 6.1 × 10^−9^ [53]. Tandem repeat genes were calculated through the physical location on the chromosome, and segmental duplication genes were defined as those located in the same synteny blocks.

### 2.5. Gene Structure and Motif Compositions Analysis

A phylogenetic tree based only 93 GmLac peptides was constructed according to same method in Section 2.3. The exon/intron organizations of soybean laccase genes were identified with the Gene Structure Display Server, GSDS 3.0 (http://gsds.cbi.pku.edu.cn/) [54]. Conserved motifs of laccase proteins were identified statistically using MEME (http://meme-suite.org/tools/meme). Discovery mode was set as discriminative mode and we used 17 *Arabidopsis* laccase protein sequences as a control. The maximum number of motifs to find was set at 10 [55]. Visualization of motif compositions was executed with TBtools V0.52 (https://github.com/CJ-Chen/TBtools).

### 2.6. Expression Pattern of Soybean Laccase Family Genes

Nine soybean-tissue/organ RNA sequencing (RNA-Seq) data, FPKM (Fragments per kilobase of transcript per million fragments mapped) values, were obtained from the Phytozome database [56]. Average expression valued were calculated, and a three-fold increase compared to the average value was defined as a high-expression gene. Color gradations (light to dark green representing different expression abundances) were appended to the expression data.

### 2.7. Expression Patterns of GmLacs Following Phytophthora sojae Infection

RNA-Seq data of Williams 82 roots 0, 0.5, 3, 6, and 12 h post *P. sojae* inoculation were obtained from the NCBI database (https://www.ncbi.nlm.nih.gov/bioproject/PRJNA318321). The variation amount of expression values (ΔFPKM) ware counted through [n_hpi (FPKM + 0.001)/0_hpi (FPKM + 0.001] and then normalized to log2 (ΔFPKM). Visualization of the expression data was conducted by the pheatmap package in R.

### 2.8. qRT-PCR Analysis of GmLacs Expression Following Phytophthora sojae Infection

*P. sojae* strain (P7076) was posted on the roots of 72-h-old seedings (grown in darkness), collecting infected samples at 0, 0.5, 3, 6, and 12 h. Total RNA of these samples was extracted using the Easy-Pure Plant RNA kit (Transgen, ER301). The RNA quality was detected through an ultra-microspectrophotometer (IMPLEN, P330). RNA samples were reverse-transcribed with EasyScript One-Step gDNA Removal and complementary DNA (cDNA) Synthesis SuperMix (Transgen, AE311) strictly following the manufacturer’s instructions. The gene expression pattern was analyzed through qRT-PCR using SYBR^®^ Green Master (Roche, 4913914001) in the Agilent Technologies Stratagene Mx3005P Detection Device. Three replicates were investigated to confirm the result. The relative expression levels were calculated via the 2^−ΔΔCt^ method after taking *GmPDF* (serine/threonine-protein phosphatase 2A regulatory subunit A gene, *Glyma.20G114000*) as an internal control. The qRT-PCR analysis primers are listed in the Appendix A.

### 2.9. Selective Pressure Analysis

SNP (Single nucleotide polymorphism) data gathered from resequencing of 302 soybean genomes were used for the genetic diversity analysis of *GmLac* genes in soybean [57]. The SNPs with missing data >10% or MAF (Minor Allele Frequency) <5% were filtered. The soybean accessions were divided into two populations: *Glycine soja* and *G. max* (landrace and improved cultivar). Nucleotide diversity (π) was calculated using a 20-k-10-k sliding window and VCFtools [58]. The ratio of diversity π_G.max_/π_G.soja_ for each window was calculated. The π_G.max_/π_G.soja_ ratio of a target region was compared to the mean π_G.max_/π_G.soja_ ratio of the corresponding entire chromosome to determine whether the target region was under selection pressure or not [59]. The 5% windows with the lowest π_G.max_/π_G.soja_ ratios values were compared with the target region π_G.max_/π_G.soja_ ratios to determine the genes that were highly selected [60].

## 3. Results

### 3.1. Identification and Classification of Laccase Genes in Soybean

To identify the laccase genes in soybean, we downloaded the reference genome of *Glycine max* from Phytozome and searched the genome with HAMMER 3.0 software for Cu-oxidase, Cu-oxidase_2, and Cu-oxidase_3 domains (PF00394, PF07731, and PF07732). After elimination of the redundant genes with only one or two typical domains on the peptide sequence or with no integral open reading frames, 93 putative soybean laccase-coding genes *GmLac1* through *GmLac93* were identified (Appendix A). Furthermore, we obtained the physical locations of these laccase genes and mapped them onto 19 of the 20 soybean chromosomes (Figure 1). Chromosome 18 had the greatest number of laccase genes, which was 12 (*GmLac70*–*GmLac81*). On the contrary, chromosomes 3, 9, 15, and 16 contained only one gene each. *GmLac91* and *GmLac92* mapped onto scaffold_27, while *GmLac93* mapped onto scaffold_614. Basic information of all soybean laccases including gene name, gene locus, amino-acid length, signal peptide length, molecular weight, pI value, and subcellular localization was also determined and is presented in the Appendix A. The amino-acid length ranged from 315 to 637 aa, with molecular weight ranging from 34,958.38 to 71,749.27 Daltons, and isoelectric points (pI) ranging from 4.95 to 9.96. Most soybean laccase pI values were above 7.00, whereas only 19 laccase pI values were below 7.00. The predicted subcellular locations of 76 soybean laccases included the extracellular space. Thirty-four of the laccases were predicted to localize only to the extracellular space. Thus, most laccases are probably secretory proteins. Laccases may also commonly be located in the plasma membrane and less commonly in lysosomes. Finally, very few laccases localized to other organelles including the mitochondria, peroxisomes, and vacuole. Specifically, *GmLac21*, *GmLac39*, and *GmLac86* may be located in the mitochondria and peroxisomes, whereas *GmLac77* may be located only in the mitochondria. *GmLac90* was the only one predicted to be located in the vacuole. Transmembrane topology and signal peptide analysis indicated that nine laccases contain transmembrane domains. *GmLac29* contains two transmembrane regions. The hydrophobic region (H-Region) of signal peptides of most laccase members ranged from 7–20 aa (Appendix A).

### 3.2. Phylogenetic Analysis of the Soybean Laccase Gene Family

To further investigate the phylogenetic relationships of laccases between soybean and *Arabidopsis*, we constructed a neighbor-joining phylogenetic tree based on multiple sequence alignments of 93 putative soybean laccase and 17 *Arabidopsis* laccase peptide sequences (Figure 2). According to classification standard of *Arabidopsis* laccases and location on the phylogenetic tree [61], 93 soybean laccases were divided into five groups. The distribution of the laccase genes in each group was rather uneven; 15 members were classified as class I, 13 members as class Ⅱ, eight members as class Ⅲ, eight members as class Ⅳ, and 49 members as class Ⅴ. Unlike in *Arabidopsis*, soybean laccase members in class Ⅳ classified into two subgroups: IVa subgroup (*Gm18* and *Gm19*) and IVb subgroup (the six other genes). None of the soybean laccases clustered into class Ⅵ. The specific gene list of each category is presented in the Appendix A.

### 3.3. Duplication and Expansion Events in Soybean Laccase Gene Family

Gene duplication is considered one of the major driving forces of genome evolution [62]. Duplicated genes are the source for creating novel genetic variation. It is generally accepted that the soybean genome went through three rounds of whole-genome duplication (WGD) events and retained more than 50% of the repeated segments [63,64]. They were the γ WGD event about ~130 to 240 million years ago (Mya), the legume WGD event about ~58 Mya, and the *Glycine* genus WGD event about ~13 Mya [65,66,67]. The general Ks rate range of soybean genome γ WGD events is bigger than 1.5, while that of the legume WGD event is 0.3 to 1.5, and that of the *Glycine* genus WGD event is 0 to 0.3 [40,44,47]. For an in-depth study the WGD events of laccase genes, we conducted collinear analysis using the base MCScanX software. In total, 85 orthologous gene pairs were identified within the pairwise syntenic blocks (Figure 3). We calculated the non-synonymous nucleotide substitution rate (Ks) and synonymous nucleotide substitution rate (Ka) between collinear gene pairs (Appendix A). We calculated that about 25.9% of gene pairs (22 of 85) were separated during the *Glycine* genus WGD event, nearly 43.5% of gene pairs (37 of 85) were separated during the legume WGD event, and approximately 30.6% of gene pairs (26 of 85) were separated during the γ WGD event. Additionally, we also calculated the Ka/Ks ratios to study the gene divergence after the WGD events. We observed that all 85 synteny pairs underwent purifying selection with Ka/Ks < 1 [48].

The most common patterns of gene duplication include interspersed repeats, tandem repeats, and segmental duplication [68], while segmental and tandem duplications are usually considered as the main causes of large gene family expansion in plants [69]. To gain insight into the soybean laccase gene duplication pattern, we analyzed the tandem and segmental duplication events of laccase genes. To identify tandem duplication, we searched adjacent gene clusters on the same chromosome with no more than one interferential gene being inserted. We found that 43.0% (40 of 93) of genes in the soybean laccase gene family were tandem repeats (Figure 1 and Appendix A).

Segmental duplications are blocks of genomic sequence ranging from 1–200 kb that map to different loci in a genome and share high sequence identity (often >90%) [70,71,72]. In this study, we defined segmental duplication as genes located on the same homologous blocks [73]. Thus, collinearity gene pairs can be regarded as segmental duplication members in the laccase gen family. In total 45.1% (42 of 93) of laccase genes involved segmental duplication (Appendix A).

### 3.4. Gene Structure and Motif Compositions of GmLacs

To reveal the structural diversity of soybean laccase genes, we constructed the exon/intron organizations and searched for conservative motifs based on the phylogenetic tree of all soybean laccase alignments (Figure 4). Overall, nearly all the closest genes on the phylogenetic tree showed remarkably similar gene structures. However, there was still a small proportion of the gene clades that exhibited different intron/exon organizations. For instance, *GmLac13* contained six exons, whereas its nearby paralogous genes, *GmLac68*, *GmLac67*, *GmLac40*, and *GmLac5*, had eight exons even though their evolutionary relationships reached 98–100% bootstrap values, respectively.

To further reveal the conserved motifs of the GmLac proteins, we analyzed 93 GmLac proteins using the MEME program and identified 10 motifs (Figure 4; Appendix A). As expected, the motif compositions of peer groups had similar structures and organizations. Some classes of GmLac proteins with completely the same motif composition may indicate the possibility of functional redundancies among those members. Some motifs were found in all 93 putative members. These conserved motifs may play critical roles in mediating GmLac function. Despite an overall similarity among the 93 GmLac proteins, there were some differences in the number of motifs. For example, the number of motifs varied from 2–10 among those GmLac proteins. Varying numbers of motifs across the GmLac proteins may suggest functional divergence among some of the GmLac proteins.

### 3.5. Expression Patterns of GmLac Genes

Gene expression patterns often reveal how, when, and where genes are active. To verify the expression profiles of *GmLac* genes, we analyzed publicly available RNA-Seq data in Phytozome. Various organs and tissues including root, root hair, nodules, pod, seed, stem, shoot apical meristem (SAM), leaves, and flower transcriptomes were downloaded (Appendix A). According to the fragments per kilobase million (FPKM) values of 90 *GmLac*s excluding the three members located in scaffolds, a differential transcriptional regulation pattern for the laccase genes was observed. *GmLac25* has the highest expression abundance and is specifically expressed in flowers (Appendix A). The overall mean FPKM value of all 90 *GmLac* genes among the nine transcriptomes was about 10. We defined the genes with mean values three-fold higher than the overall mean FPKM value across all transcriptomes as highly expressed *GmLac* genes. With this criterion, we identified 11 genes that were highly expressed in multiple tissues (Table 1). Most of the highly expressed genes showed transcript abundance in the stem, indicating that they may be involved in stem cell-wall maintenance to provide mechanical strength for the aboveground portion of the plant. Conversely, the expression of some genes was restricted to a specific organ or tissue and very poorly expressed in others. For example, *GmLac1* and *GmLac24* specifically and highly expressed in the roots, *GmLac77* did so in pods, *GmLac2*, *GmLac8*, and *GmLac9* did so in the stem, *GmLac87* did so in the SAM, and *GmLac25* did so in the flower (Table 2). The promoters of these genes were active in specific organs or tissues. In addition, 11 genes were inactive in all tissues and organs studied (Appendix A). For the remaining members, there was either a mild expression pattern with moderate abundance or expression in several samples.

We also compared the expression pattern of collinear gene pairs (Appendix A). In total, 64.7% (55 of 85) of collinear gene pair expression patterns were distinct. Only a fraction of gene pairs (23 of 85) maintained a similar expression pattern. These results additionally suggested a differentiation of the promoter elements of laccase genes and possibly their functions.

### 3.6. Expression Changes of GmLacs after Inoculation with Phytophthora sojae

It is well established that laccases are involved in lignin metabolism and plant defenses [34,35,36,74,75]. *Phytophthora sojae* is an oomycete pathogen that causes root and stem rot disease in soybean, which severely affects yield [76]. We were interested if any laccase genes were rapidly induced following *P. sojae* infection. An earlier transcriptomic dataset of soybean–*P. sojae* interaction is available on the NCBI database (https://www.ncbi.nlm.nih.gov/bioproject/PRJNA318321). We studied the responses of 93 soybean laccase genes following infection (Appendix A). Considering 0 h post infection (hpi) as a control, we calculated the expression change value of the *GmLac* genes at 0.5, 3, 6, and 12 hpi (Appendix A), and the results are depicted in a heatmap (Figure 5). According to the clustering, 93 putative laccase genes can be roughly divided into three categories based their expression status. Most laccase genes (72 of 93) did not show obvious expression changes following *P. sojae* infection. Nine of the remaining were induced, including *Gmlac28*, *Gmlac91*, *Gmlac92*, *Gmlac79*, *Gmlac23*, *Gmlac57*, *Gmlac47*, *Gmlac37*, and *Gmlac36*. Although they were induced, their expression levels peaked at different stages. Expression abundances of the remaining 12 genes were suppressed. *GmLac32* and *GmLac75* were strongly repressed at 12 hpi, while the other nine (*GmLac7*8, *GmLac7*1, *GmLac25*, *GmLac89*, *GmLac15*, *GmLac90*, *GmLac45*, *GmLac48*, *GmLac7*6, and *GmLac18*) were potentially suppressed at multiple stages. By examining the expression profile of these 21 genes (Appendix A), we noticed that these genes were barely expressed in normal conditions. On the other hand, after infection, the expression of these genes changed significantly, which strongly implies a close tie between these genes and oomycete pathogen resistance.

To further test the expression pattern of these genes, we performed qRT-PCR on a Williams 82 root sample with a soybean *Phytophthora sojae* strain (P7076) [77]. The results (Figure 5) suggested that most (8/9) induced laccase genes were similarly induced with the exception of *GmLac36*. In particular, *GmLac28*, *GmLac91*, *GmLac92*, *GmLac79*, *GmLac23*, and *GmLac57* expression abundances were significantly increased in both tests. Conversely, the quantitative results of repressed genes also displayed coincident RNA-seq data. *GmLac78*, *GmLac75*, *GmLac32*, *GmLac32*, *GmLac45*, *GmLac89*, and *GmLac25* showed obvious repression in expression. *GmLac48* showed no expression signal. Automatically, we suggest a putative defense function for the *P. sojae*-induced genes, while suppressed genes could equally be important, as *P. sojae* effectors may manage to suppress the expression of some defense-related *GmLac* genes to promote colonization.

### 3.7. Selection Pressure Analysis of Laccase Gene Family during Soybean Domestication

Cultivated soybean was domesticated from its relative *Glycine soja* in ancient China [78], which led to a series of “domestication syndromes” and subsequently resulted in markedly decreased nucleotide diversity (π) of the respective domestication genes [79]. Wild soybeans uniformly exhibit a softer and thinner stem, while cultivated soybeans uniformly show thicker, stronger, and erect stems, suggesting that these phenotypes are most likely domestication-related traits [80].

Lignin is abundant in the stem cortex and xylem cell wall, playing an important role in maintaining mechanical strength of stem [81]. Laccase is well known for its ability to facilitate lignin synthesis and promote plant tissue-specific lignification [82]. Therefore, we wanted to investigate whether the domestication of soybean was accompanied by changes in laccase gene nucleotide diversity. Thus, we performed selection pressure analysis of *GmLac* genes based on π ratio (π*_G.max_*/π*_G.soja_*) between 62 wild species and 240 (130 landraces and 110 improved cultivars) cultivated soybean accessions [57]. A comparison of the π*_G.max_*/π*_G.soja_* values of a chromosomal region containing *GmLac* genes with the corresponding mean value of the respective chromosome (Appendix A) revealed that 57 of 90 (63.3%) *GmLac* genes exhibited above average π*_G.max_*/π*_G.soja_* values. This observation indicates that the nucleic acid diversity of *GmLac* genes significantly decreased following the domestication of soybean, and also suggests that most laccase genes were under selective pressure during the domestication of cultivated soybean (Appendix A). When judging whether a gene underwent selection pressure by comparing with the top 5% of π*_G.max_*/π*_G.soja_* values according to previous research [57,60], we identified 10 genes located in the selective sweep regions (Figure 6; Appendix A). Of these 10 genes, *GmLac8*, *GmLac12*, *GmLac66*, and *GmLac85* are highly expressed in the stem, and, more importantly, the expression of *GmLac8* is highly specific to the stem (Appendix A). These genes are most likely involved in the transition from a soft and thin soybean stem to a thick and erect stem.

To further support our conclusion that 10 genes have underwent strong selection pressure during the domestication of soybean, we conducted phylogenetic analysis using SNPs of the ten genes among 302 resequenced soybean accessions (Appendix A). The 302 accessions can be divided into five groups (Figure 7). In Groups Ⅰ through Ⅳ, all accessions except SRR1533282 are wild soybean species. In Group Ⅴ, all but SRR1533169 and SRR1533198 are soybean cultivars. These results also imply that the nucleotide diversity of the 10 genes is much higher in the wild accessions than in the cultivated accessions.

## 4. Discussion

Laccase enzymes are multifunctional proteins. They were shown to contribute toward cell morphology [32,33], pigment formation [83], secondary cell-wall biosynthesis [84,85], and resistance to biotic and abiotic stresses in plant [74,75]. Functional studies of laccases were already carried out in many model and crop plants, including *Arabidopsis* [86], *Brachypodium distachyon* [87], sorghum [42], *Oryza sativa* [43], *Populus trichocarpa* [88], *Gossypium raimondii*, and *Gossypium arboretum* [74]. However, as one of the most important crops around the world, the functions of soybean laccase are rarely reported. In this study, gene structures, protein properties, phylogenetic relationships, replication characteristics, expression profiles, and selection pressures were investigated for members of the soybean laccase gene family.

The soybean Williams 82 genome encompasses 105 putative copper oxidase domain-containing genes that can be regarded as putative candidate laccase genes. Typical laccases usually have three copper oxidase domains; therefore, we investigated 93 *GmLac* genes that contained three copper oxidase domains, representing 0.2% of the total number of soybean genes, which is higher than the laccase gene proportions in other plant species including *Arabidopsis thaliana* (0.06%) [89], *Brachypodium distachyon* (0.13%) [90], *Sorghum bicolor* (0.08%) [91], *Oryza sativa* (0.08%) [92], *Populus trichocarpa* (0.07%) [93], *Gossypium raimondii* (0.11%) [94], and *Gossypium arboretum* (0.11%) [95]. Predicted isoelectric points of most soybean laccases (74 of 93) are higher than 7, suggesting that they tend to function in alkaline environments. Predicted subcellular locations of the GmLac proteins suggest that the vast majority of laccases (76 of 93) are extracellular. This result implies that most soybean laccases are secretory proteins.

Through multiple sequence alignment of 17 *Arabidopsis* and 93 soybean laccases and construction of an N-J phylogenetic tree, we divided the soybean laccases into five classes according to the classification criteria applied in *Arabidopsis thaliana*. Class Ⅳ was divided into two different branches, Ⅳa and Ⅳb. Further genome-wide duplication (GWD), homologous, and historical duplication event analyses of the *GmLac* genes revealed that laccase genes are an ancient gene family that expanded as early as ~321 Mya ago during the γ WGD event that was accompanied by the evolution of rosids. Laccase genes continued to expand in subsequent legume WGD and *Glycine* WGD events. According to the Ks value, purifying selection was the major force driving the evolution of the laccase gene family. Synteny analysis and the physical location of *GmLac* genes show that tandem repeats (41.93%) and segmental duplications (43.01%) were the main mechanisms of laccase gene duplication.

Gene structure and protein motif analyses showed that the most closely related members in the phylogenetic tree had similar exon/intron organization and common motif compositions. This may suggest the existence of a possible functional similarities among the closely related GmLac proteins. However, within the five basic classes, exon/intron organization and motif compositions were highly diverse, suggesting the occurrence of functional differentiation in each subgroup of the soybean laccase gene family.

Investigation of the expression profiles of *GmLac* genes among nine organs/tissues revealed the possible role of transcriptional regulation in mediating the functions of GmLac proteins. Some members have high expression abundance, while some genes are barely expressed. Both organ/tissue-specific and constitutive types of expressions were observed among members of the gene family.

In response to *P. sojae* infection, only a few *GmLac* genes were either induced or repressed. Although it is possible to assign a putative defense function for the *P. sojae*-induced genes, downregulation of some genes could equally be important in *Phytophthora* resistance. Exchange of the promoter of such a gene was shown to enhance resistance in transgenic soybean plants against this pathogen [96]. *P. sojae* effectors may manage to suppress the expression of some defense-related *GmLac* genes to avoid lignified cell barriers. Laccase genes are able to enhance plant resistance through accelerating the polymerization of monolignol to lignin. On one hand, increased lignin content helps create stronger physical barriers to prevent invasion; on the other hand, lignin synthesis reinforces cell biosynthesis, which is a branch of the phenylpropyl pathway, and phenylpropanoid metabolism is one of the most important secondary metabolic pathways involved in plant defense against biotic and abiotic stresses [75].

Our study on the natural variations in the *GmLac* gene family across a collection of both wild and cultivated accessions revealed that the laccase gene family may have been subjected to selection pressure following domestication (Figure 6). Together with the differential expression patterns and selection pressure, four genes were found to be highly expressed in the stem. Based on the differences in stem characteristics between wild soybean and cultivated soybean and laccase’s role in lignification, we speculate that these four genes may have made an important contribution to the transition from a soft and thin stem in wild species to a hard, thick, and erect stem in present day soybean cultivars.

## 5. Conclusions

A total of 93 putative *GmLac* genes were identified from the William 82 genome. Based on the phylogenetic relationships between these genes and *Arabidopsis thaliana* laccase genes, they were divided into five classes (Figure 1 and Figure 2; Appendix A). Gene duplication and expansion analysis showed that, as expected, duplication of the members of the soybean laccase gene family occurred in all three WGD events in soybean (Figure 3; Appendix A). Purifying selection was the major driving force in laccase gene family evolution. The main gene duplication mechanisms for *GmLac* gene family were tandem and segmental duplications. The analyses of conserved domains suggested that, in general, adjacent members in the phylogenetic tree had common motif compositions. However, exceptions were observed within a clade for motif compositions. Differences were recorded for functional conservation, as well as in the motif compositions within clades (Figure 4; Appendix A). Transcriptomic analysis revealed both tissue- and organ-specific specific expression patterns, as well as constitutive expression patterns, of the soybean laccase gene family (Table 1 and Table 2; Appendix A). This study also revealed that distinct *GmLac* gene sets are induced or repressed in response to invasions by oomycete pathogen *P. sojae* (Figure 5; Appendix A). Selective pressure analysis revealed a possible role of a few members of the gene family in transitioning from the soft and thin stem of wild soybean species to the strong, thick, and erect stem of the cultivated soybean accessions.

## Figures and Tables

**Figure 1 genes-10-00701-f001:**
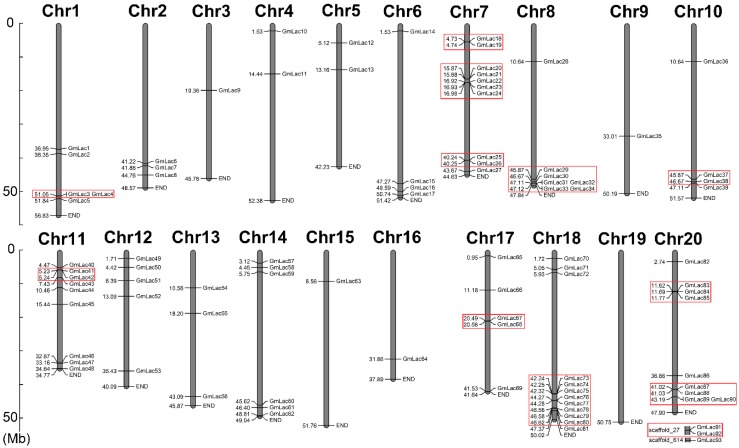
Physical map of the soybean laccase genes. Chromosome size is indicated by its relative length. The scale bar is shown on the left, and the numbers represent the physical position of the genes on the chromosomes. Tandemly duplicated genes are represented by boxes with red outlines.

**Figure 2 genes-10-00701-f002:**
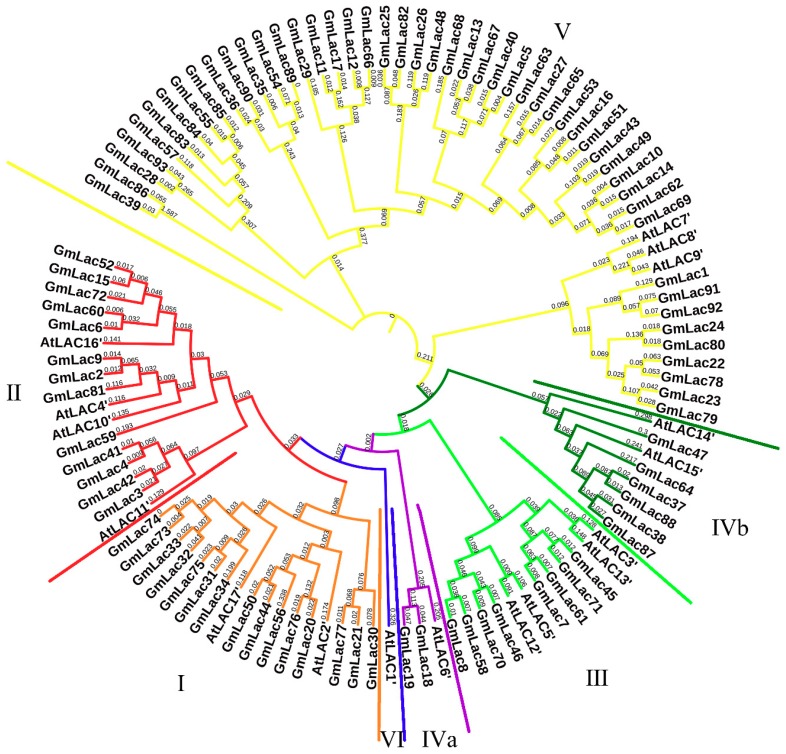
The phylogenetic tree of the *Arabidopsis* and soybean laccase proteins. In total, 93 soybean and 17 *Arabidopsis* laccase proteins were aligned using the ClustalW program. The neighbor-joining tree was constructed using MEGA7.0. The bootstrap value was set as 1000 replicates. The length of each branch is shown next to the branches.

**Figure 3 genes-10-00701-f003:**
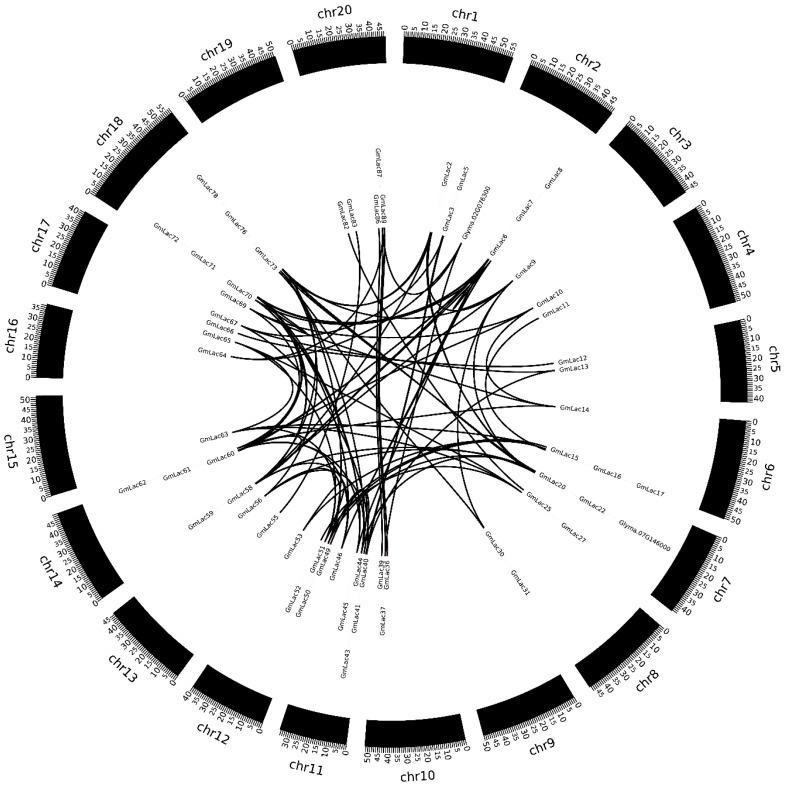
Collinearity of the laccase gene (*GmLac*) gene pairs. The outermost scale represents the chromosomes. Genes are listed in the inner circle according to their chromosomal location. Synteny relationships between gene pairs are marked with a black line.

**Figure 4 genes-10-00701-f004:**
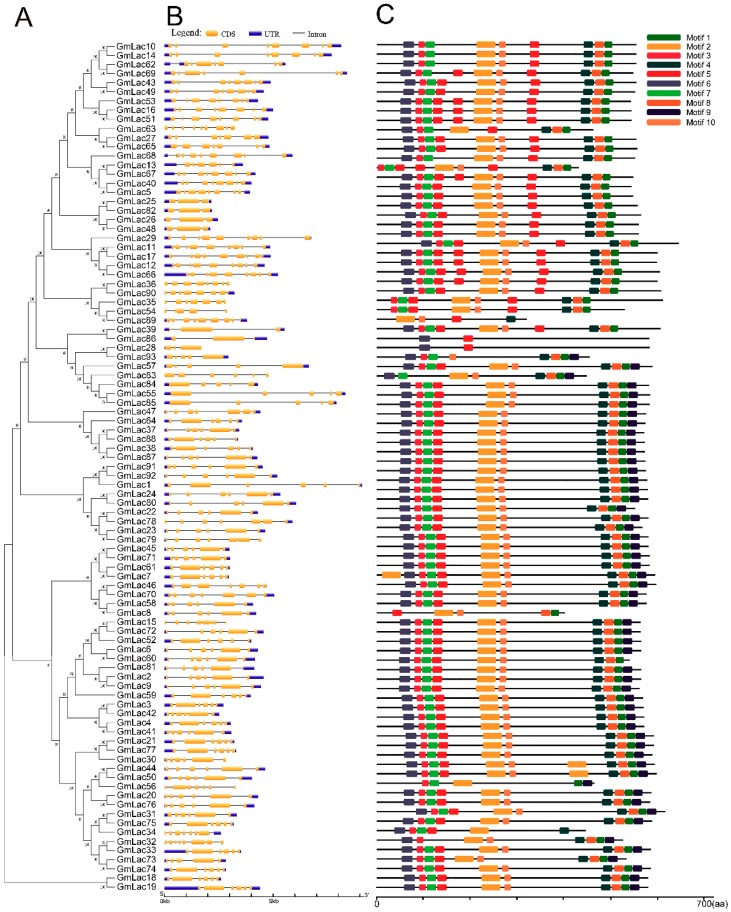
Phylogenetic relationships, gene structure, and motif compositions of the soybean laccase gene family. (**A**) A neighbor-joining tree of 93 GmLac protein sequences constructed using MEGA v7.0. (**B**) The structure of the 93 *GmLac* genes. Yellow squares correspond to exons and linking black lines indicate introns, while the blue squares refer to the 5′ untranslated region (UTR) and 3′ UTR sequences. (**C**) Schematic motif composition of 93 GmLac proteins. The colored boxes represent the different motifs, indicated in the top right-hand corner. The scales at the bottom of the image indicate the estimated exon/intron length in kb and motif length in numbers of amino acids (aa).

**Figure 5 genes-10-00701-f005:**
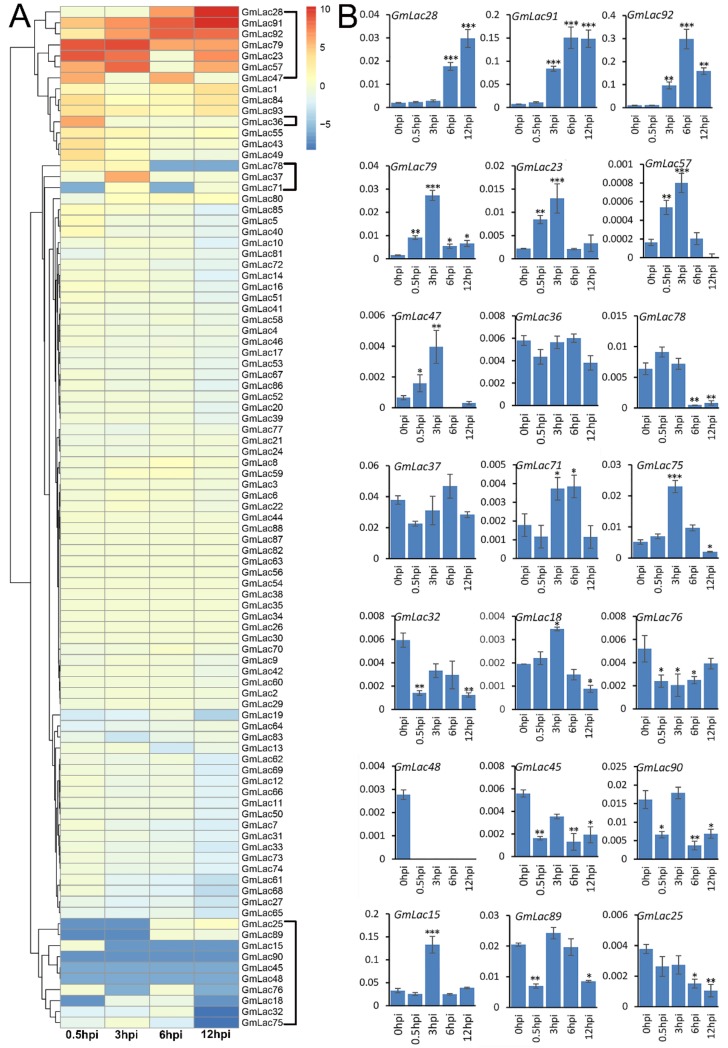
Expression profiles of the 93 soybean laccase genes following *Phytophthora sojae* infection. (**A**) Transcriptome heatmap. The hierarchical cluster color code: the highest induction values vs. controls are displayed in red (hot), while the values displayed in blue (cool) indicate the most reduced expression levels vs. controls; intermediate expression levels are represented by lighter intermediate colors. The color scale represents △FPKM normalized to log2-transformed vales. The genes marked by the black brackets on the right in (**A**) are those obviously mis-regulated genes. (**B**) Expression levels of 21 *GmLac*s after P7076 infection. Bars represent average values of three replicates ± standard deviation (SD). All expression levels of the *GmLac* genes were normalized to the expression levels of *GmPDF*. ***, **, and * indicate significant differences compared to the control (0 h) at *p* < 0.001, *p* < 0.01, and *p* < 0.05, respectively.

**Figure 6 genes-10-00701-f006:**
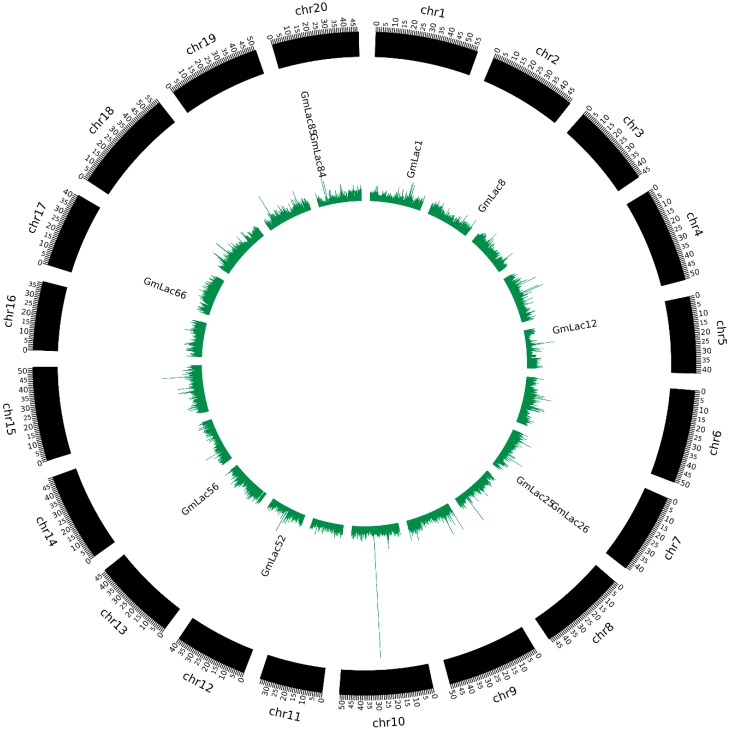
Selective pressure analysis of laccase gene family. The outermost scale represents chromosomes, and the green bar chart in the middle represents π*_G.max_*/π*_G.soja_* values per chromosome. The top 5% selected laccase genes are listed according to their physical location.

**Figure 7 genes-10-00701-f007:**
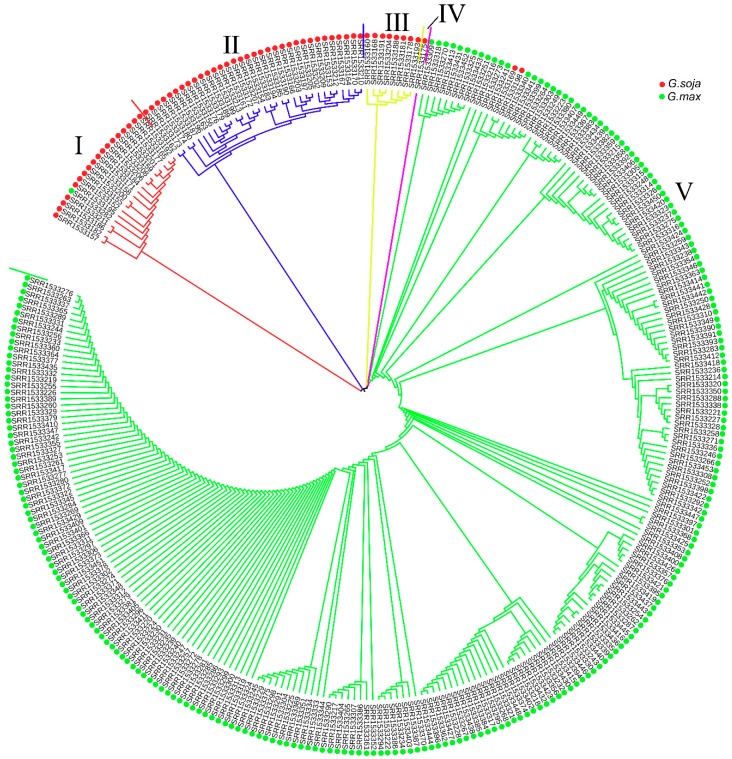
Phylogenetic tree of 302 accessions. SNPs of the 10 selected genes among resequenced genomes of the 302 soybean accessions were used to construct the neighbor-joining (N-J) phylogenetic tree. The bootstrap value was set as 1000 replicates. Corresponding specie idnetifiers (IDs) of the accessions are available from Reference [57].

**Table 1 genes-10-00701-t001:** Constitutively expressed laccase genes (*GmLac*).

Gene Name	Gene ID	Root	RH	Nodules	Pod	Seed	Stem	SAM	Leaves	Flower	Scale Bar
*GmLac5*	*Glyma.01G183100*	29.87	66.04	33.99	37.23	14.55	46.41	24.14	2.00	6.78	0.00
*GmLac10*	*Glyma.04G019500*	11.59	44.89	11.61	73.40	111.75	53.38	23.71	2.26	11.45	5.00
*GmLac12*	*Glyma.05G056100*	54.73	65.77	60.52	23.36	22.16	47.63	55.08	6.07	5.52	10.00
*GmLac16*	*Glyma.06G307100*	36.50	49.47	10.29	129.80	4.81	99.05	114.40	14.28	18.43	15.00
*GmLac40*	*Glyma.11G059200*	33.53	59.28	40.51	33.92	10.26	87.08	43.94	6.10	9.01	20.00
*GmLac55*	*Glyma.13G076900*	162.50	50.75	30.23	12.38	3.09	36.75	8.77	12.19	4.63	25.00
*GmLac62*	*Glyma.14G223000*	27.32	53.26	32.22	21.12	15.68	33.04	8.42	1.00	2.10	30.00
*GmLac65*	*Glyma.17G012300*	69.51	127.58	54.73	26.11	33.73	26.31	62.91	1.39	4.16	
*GmLac66*	*Glyma.17G138300*	66.83	56.05	48.36	56.52	35.88	84.13	43.27	8.40	9.79	
*GmLac69*	*Glyma.17G261500*	47.09	65.88	29.72	2.73	0.36	28.53	16.04	3.85	4.14	
*GmLac85*	*Glyma.20G051900*	93.65	67.15	30.28	32.47	44.21	48.51	13.49	12.04	2.55	

Note: ID, identifier; RH, root hairs; SAM, stem apical meristem. Data in the table are FPKM (fragments per kilobase million) values obtained from Phytozome. Table colors are labeled according to scale bar.

**Table 2 genes-10-00701-t002:** Organ-/tissue-specific expression of *GmLac* genes.

Gene Name	Gene ID	Root	RH	Nodules	Pod	Seed	Stem	SAM	Leaves	Flower	Scale Bar
*GmLac1*	*Glyma.01G108200*	41.64	4.94	6.82	0.06	0.00	0.04	0.01	0.03	0.14	0.00
*GmLac2*	*Glyma.01G112600*	0.94	6.30	12.58	10.45	0.45	70.53	3.31	5.93	1.92	5.00
*GmLac8*	*Glyma.02G261600*	2.95	4.36	3.94	7.97	0.69	32.58	0.56	0.98	1.91	10.00
*GmLac9*	*Glyma.03G077900*	1.00	7.28	15.93	10.78	0.61	75.27	3.25	5.62	2.18	15.00
*GmLac24*	*Glyma.07G142600*	38.53	11.66	21.13	0.08	0.00	0.03	0.09	2.86	0.21	20.00
*GmLac25*	*Glyma.07G225300*	0.20	0.00	0.00	0.01	0.00	1.06	0.20	0.03	505.03	25.00
*GmLac77*	*Glyma.18G183700*	0.16	1.65	2.13	35.66	0.05	16.90	0.30	1.16	0.32	30.00
*GmLac87*	*Glyma.20G172600*	13.60	0.03	0.00	0.00	0.01	2.87	117.91	0.00	2.54	

Note: ID, identifier; RH, root hairs; SAM, stem apical meristem. Data in the table are FPKM (fragments per kilobase million) value obtained from Phytozome. Table colors are labeled according to scale bar.

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
