# Peer review of "The Soybean Laccase Gene Family: Evolution and Possible Roles in Plant Defense and Stem Strength Selection"

_genes, 2019, doi:10.3390/genes10090701_

Round 1
Reviewer 1 Report
Check minor spelling mistakes, I highlighted them in the PDF

Reviewer 2 Report
Authors present a research article entitled: "The Evolution and Role of Soybean Laccase Gene Family in Plant Defense and Stem Strength Selection".
First of all, the authors should use the service of native English speaker in order to correct their manuscript and enrich it as well. I feel that especially the result and discussion part are suffering from poor English and phrasing. The description of the results is often badly done.
The title does not reflect the results. Indeed, authors identified 93 putative soybean laccases. using bioinformatics tools (phylogenetic tree, conserved domains and localization prediction, etc.), they characterized this protein family. Using publicly available RNA-seq data, authors tried to show how the expression of this gene family is changed during infection by 2 pathogens. By the way, this part of the results is badly described and it is hard to make any conclusions.
Based on ONLY bioinformatic analysis authors want to ascribe a role to these laccases during plant defense and stem strength. However, they did not validate their conclusion by at least some qPCR analysis, or the use of mutants, or other ways. Therefore, from my opinion the data provided in the manuscript support the "identification and evolution of soybean laccase" but do not support their "role in defense or/and stem strength".
I suggest either to change the title or make the experiments that would support their title.
Some parts of the results belong more to the discussion, or introduction. For example:
"It is well established that laccases are involved in lignin metabolism and plant defenses [34-36,75]. Polymerization of monolignol to lignin provides mechanical strength and reinforces cell biosynthesis is a branch of phenylpropyl pathway while phenylpropanoid metabolism is one of the most important secondary metabolic pathway involved in plant defense against biotic and abiotic stresses". This can be found in the result section, but such information would be of greater interest in the introduction as a justification of the study, and also in the discussion. For instance, in the discussion related to resistance to infection, the authors could have named the laccases for which the expression was induced upon infection. Here they could then suggest that the corresponding enzymes could be important in lignin synthesis, creating a physical barrier against pathogen and try to identify whether these soybean laccase are homologs to (Arabidopsis) laccases known to have a role in such response (i.e. infection).
While proceeding in this way, authors could have a stronger article,well discussed... which is not the case with the version provided for reviewing.
I'm adding here a pdf of the manuscript where authors can see all my concerns related to the manuscript.
I can only provide the following recommendation: "RECONSIDER AFTER MAJOR REVISION", not because some experiments are missing, but because authors have to provide an improved manuscript. I believe that the topic is interesting, but unfortunately the work is not accomplished.

Reviewer 3 Report
It's an interesting elaborate study of an important gene family that requires attention in the soybean research community. The results presented in this manuscript provide a strong foundation for future studies on the laccase enzymes in soybean.
I actually used the manuscript to edit and write my comments which can easily be fixed. The edited manuscript is uploaded. Please look for yellow highlight or colored fonts of red or blue color for comments or changes. Title has also been edited.
Authors may consider to carefully check the manuscript for the suggested edits as well as for the references and other supplemental documents, citation of figure and table numbers in the text carefully.
May consider to add putative in front of GmLac genes in additional appropriate places because the gene family has not been characterized for their function. Prediction is based on only sequence comparison.
